# OmniVIVO: Towards Unified Multimodal Generative Modeling for Simultaneous Language-Guided Speech and Image Synthesis

## Abstract

Recent large language models (LLMs) based on autoregressive (AR) next-token prediction have achieved remarkable success in natural language generation and are rapidly expanding to image and speech synthesis. Yet most current approaches still treat these modalities in isolation—training independent models or loosely coupling multiple generators. Even recent omni-models such as UGen and Qwen2.5-Omni mainly address understanding tasks or text–image generation and do not provide a single AR backbone capable of simultaneously producing high-fidelity images and natural speech. Inspired by the human brain's capability to imagine and speak simultaneously, we propose **OmniVIVO**, a unified AR approach for modeling **vi**sual and **vo**ice modalities together, capable of generating high-fidelity images and natural speech in parallel from a single text input. Our OmniVIVO integrates a state-of-the-art AR image generator with a novel lightweight speech decoder, enabling the first unified approach for the concurrent generation of natural speech and high-fidelity images. By sharing representations across modalities within a single transformer backbone, the model learns a rich multimodal space that enables tighter semantic alignment and more efficient joint generation than existing multi-model pipelines. Through a unified backbone, OmniVIVO produces speech with high perceptual quality and naturalness, surpassing comparably sized text-to-speech (TTS) systems and being on par with state-of-the-art systems like Cosyvoice2 and VITS, while maintaining high-fidelity image generation. To quantify contextual understanding across modalities, we propose a **multimodal ranking metric** spanning text, speech, and images, demonstrating that OmniVIVO's bi-modal outputs are effective in information acquisition. We construct **VIVOGen**, a high-quality tri-modal text–image–speech dataset that leverages OmniVIVO's multimodal outputs, providing a valuable resource for research in multimodal learning and applications in education and language acquisition, which we will publicly release.

## 1 Introduction

Recent advancements in artificial intelligence (AI), driven by large language models (LLMs) such as GPT-4 (OpenAI, 2023), LLaMA-3 (Dubey et al., 2024), and Qwen2.5 (Yang et al., 2024), have significantly expanded the boundaries of AI, making powerful tools more accessible to a broader audience. Recently, research has expanded LLM applications to new areas such as visual generation and speech synthesis. Noteworthy achievements have been made in text-to-image (T2I) generation, where LLMs produce high-fidelity images, rivaling the performance of task-specific models such as diffusion-based architectures (Sun et al., 2024; Tian et al.). In the domain of speech synthesis, LLMs have also demonstrated impressive progress in text-to-speech (TTS) systems, generating highly natural-sounding speech (Gao et al., 2025) while showcasing advanced capabilities such as prompt-based control and zero-shot learning (Du et al., 2024). Furthermore, recent studies have explored multimodal LLMs such as Qwen2.5-Omni (Jin Xu, 2025), UGen (Tang et al., 2025), which are capable of processing inputs from diverse modalities (images, text, and speech) and producing text or speech output.

While much research Bai et al. (2023); Tang et al. (2025) in multimodal AI has primarily focused on processing and understanding multimodal inputs, some recent efforts Team (2024); Lu et al. (2023) explore multi-output generation but still lack the ability to produce speech. In particular, producing outputs such as speech and images is a fundamental step toward emulating the human capacity to imagine and vocalize simultaneously. To bridge this gap, we propose a novel approach that enables the concurrent generation of high-fidelity images and natural-sounding speech from a unified LLM architecture. This method not only mimics human-like cognition but also enhances the flexibility and scalability of multimodal systems, paving the way for more immersive human-AI interactions, including dynamic storytelling applications.

We present **OmniVIVO**, the first model capable of generating both **vi**sual and **vo**ice outputs within a unified LLM backbone. Unlike prior works that treat image and speech generation as independent tasks, OmniVIVO integrates a state-of-the-art (SOTA) pretrained image generator with a novel lightweight yet effective speech generation module, enabling parallel production of both high-quality images and speech. Through this unified design, experiments show that OmniVIVO not only surpasses TTS baselines of comparable scale in speech quality but also demonstrates strong cross-modal coherence, marking a step toward truly multimodal generative intelligence.

To evaluate multimodal generation, we further introduce a ranking metric designed to assess the subjective quality of OmniVIVO's outputs. Experimental results confirm that OmniVIVO produces multimodal outputs that are highly effective for information acquisition, highlighting its promise for diverse applications. In addition, we release **VIVOGen**, a high-quality tri-modal dataset consisting of language-specific knowledge inputs paired with OmniVIVO-generated image and speech outputs, fostering progress in multimodal education, language acquisition, and related domains.

The key contributions of this work include: (a) we introduce **OmniVIVO**, the first unified LLM model capable of simultaneously generating high-quality speech and images; (b) we present a new **multimodal ranking metric** to evaluate the quality of OmniVIVO's outputs and multimodal generation; and (c) we release **VIVOGen**, a high-quality tri-modal text–image–speech dataset designed to advance research in multimodal applications.

## 2 RELATED WORK

**Autoregressive Image Generation:** Early advancements in image generation transitioned from GANs (Goodfellow et al., 2020), which achieved high fidelity but struggled with unstable training, to diffusion models (Ho et al., 2020), which have become the dominant approach due to their scalable architectures and stable likelihood-based training. However, diffusion models are computationally expensive during inference due to the iterative denoising process, which has sparked renewed interest in autoregressive (AR) approaches that directly model discrete image tokens in sequence. Initial AR models, such as VQ-VAE (Van Den Oord et al., 2017) and VQGAN (Esser et al.), employed transformer decoders to predict image tokens in a raster-scan order. While these models demonstrated feasibility, they were inefficient and produced spatially unnatural results. The introduction of VAR (Tian et al.), which employed a coarse-to-fine next-scale prediction strategy, improved image quality but still required thousands of tokens per image, resulting in significant computational costs.

Recent advances have redefined AR generation by leveraging the power of LLMs. For instance, LlamaGen (Sun et al., 2024) demonstrates that scaling a vanilla decoder-only LLM to billions of parameters, combined with carefully curated data, enables AR models to match or even surpass diffusion models on ImageNet (Deng et al., 2009). Similarly, Muse (Chang et al.) employs masked AR training with LLM-based techniques to achieve T2I quality on par with diffusion models.

**Autoregressive Text-Speech Language Model:** Modern TTS systems have recently made a breakthrough by moving from specialized modules such as Tacotron2 (Shen et al., 2018) and FastSpeech2 (Ren et al.) to architectures based on LLMs (Du et al., 2024). This shift enables models to leverage the power of pretrained LLMs trained on massive datasets, thereby improving contextual modeling and producing speech that is natural, expressive, and high-fidelity. Instead of separating linguistic and acoustic processing, LLM-based TTS unifies the workflow by modeling sequences of discrete units (semantic, prosodic, acoustic) quantized from speech signals, thus providing a smoother bridge between text and speech (Wang et al.).

Pioneering models that applied LLMs to speech generation include VALL-E (Wang et al.), which demonstrated zero-shot TTS from just a few seconds of reference audio. More recently, systems such as CosyVoice2 (Du et al., 2024) and Spark-TTS (Wang et al., 2025) have advanced this direction further, not only achieving high synthesis quality but also supporting advanced controllability features such as instruction prompting, zero-shot. These advances mark a paradigm shift from task-specific pipelines toward general-purpose generative models, where a single backbone can flexibly handle speech generation.

**Multimodal Large Language Models:** Recent advancements in multimodal large language models (MLLMs) have extended the capabilities of text-based models to process different modalities. Systems like Flamingo (Alayrac et al.), Qwen2.5-Omni (Jin Xu, 2025), UGen (Tang et al., 2025) integrate vision and speech with text, enabling models to process diverse inputs. However, these approaches are primarily focused on multimodal understanding, where inputs come from various sources, but the outputs are typically limited to text or speech.

Recent work by (Team, 2024) presents a multimodal large language model (MLLM) capable of bidirectional generation for both text and images. This capability is enabled by a unified autoregressive backbone trained alongside an image tokenizer and a text tokenizer, supporting multiple input–output configurations. Unified-IO (Lu et al., 2023) similarly adopts a single transformer architecture to address diverse computer vision and vision–language tasks through a unified discrete representation. However, these works (Team, 2024; Lu et al., 2023) do not extend to the generation of speech outputs, which limits their applicability in broader multimodal settings. MingOmni (AI, 2025) attempts to support a wider range of modalities, including speech, yet still relies on task-specific generators such as diffusion models for image synthesis. This dependency adds extra complexity and prevents the model from being truly unified.

To address this gap, we propose OmniVIVO, the first unified LLM model designed to generate both images and speech simultaneously from a single text input. By leveraging a SOTA image generator with a lightweight speech decoder under a shared LLM backbone, OmniVIVO pushes the boundaries of multimodal generation, enabling groundbreaking advancements in fields such as multimodal education, language acquisition, and interactive AI.

## 3 METHODOLOGY

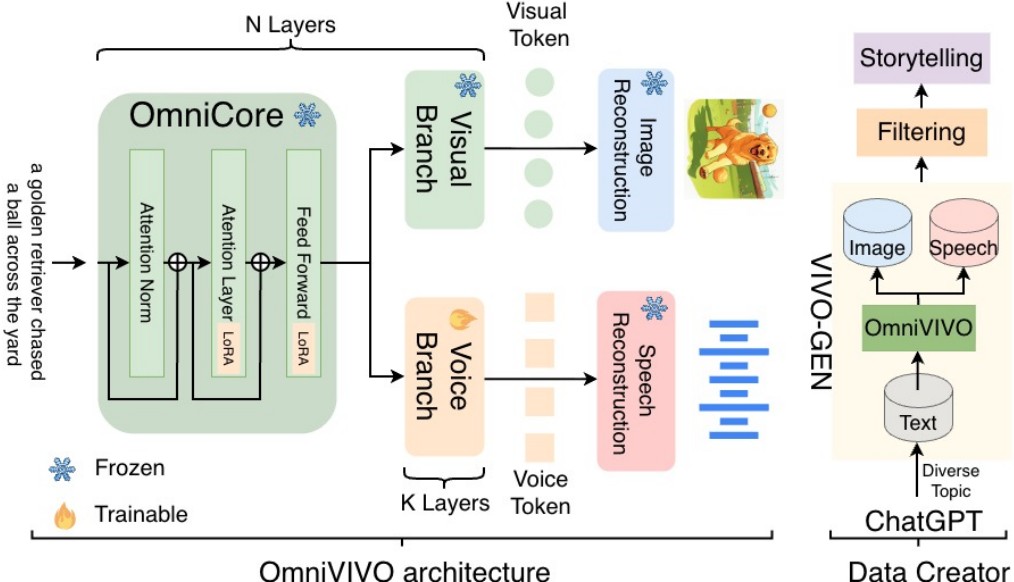

Figure 1: OmniVIVO: the proposed model is capable of generating high-quality images and speech. Additionally, VIVOGen is released to advance multimodal applications like storytelling.

### 3.1 PROPOSED MODEL: OMNIVIVO

We propose **OmniVIVO**, shown in Figure 1, a multimodal LLM model capable of generating both images and speech from a text input. Our design leverages a pretrained image backbone as a shared semantic encoder, augmented with lightweight LoRA adapters and an additional speech branch. This approach preserves image generation quality while enabling efficient cross-modal adaptation.

#### 3.1.1 OMNICORE

First, the input text is tokenized by the pretrained Flan-T5 tokenizer (Chung et al.) into a vector $x$ with $T$ tokens:

$$x = (x_1, \ldots, x_T), \quad x_t \in \mathcal{V}_{\text{text}}.$$

We adopt a pretrained image generator backbone $f_\theta$ (LlamaGen (Sun et al., 2024)) as the shared transformer. To enable adaptation without degrading image generation, we freeze all original parameters $W_\theta$ and insert low-rank adapters (LoRA Hu et al. (2022)) into each linear projection:

$$W_\beta = W_\theta + \alpha \cdot W_A W_B,$$

where $W_A \in \mathbb{R}^{d_{\text{in}} \times r}$ and $W_B \in \mathbb{R}^{r \times d_{\text{out}}}$ are trainable low-rank matrices, and $\alpha$ is a fixed scaling factor. This yields an adapted backbone $f_\beta$ that retains the representational capacity of $f_\theta$ while gaining flexibility for new modalities.

#### 3.1.2 IMAGE GENERATION

For image generation, the token vector $x$ is embedded and passed through the frozen backbone to obtain the latent feature $h_i$ with $d$-dimension:

$$h_i = f_\theta(x), \quad h_i \in \mathbb{R}^{T \times d}.$$

The pretrained output head $W_{\bar{\theta}}$ from LlamaGen (Sun et al., 2024) is kept fixed. The conditional distribution over $N$ image tokens $\bar{E}$ is:

$$p_\theta(\bar{e} \mid x) = \prod_{n=1}^{N} \text{Softmax}(W_{\bar{\theta}} h_i[n]).$$

Because neither $f_\theta$ nor $W_{\bar{\theta}}$ is updated, OmniVIVO can maintain the image generation performance of the pretrained model.

#### 3.1.3 SPEECH GENERATION

To extend the backbone for speech generation, we attach a lightweight speech transformer $f_\phi$ on top of the adapted backbone to obtain the latent feature $z_s$:

$$h_s = f_\beta(x),$$

$$z_s = f_\phi(h_s), \quad z_s \in \mathbb{R}^{T \times d}.$$

The outputs are projected into $M$ discrete speech tokens $\bar{S}$ through a trainable head $W_{\bar{\phi}}$:

$$p_\phi(\bar{s} \mid x) = \prod_{m=1}^{M} \text{Softmax}(W_{\bar{\phi}} z_s[m])$$

Here, $\{W_A, W_B, f_\phi, W_{\bar{\phi}}\}$ are updated during training.

#### 3.1.4 OMNIVIVO TRAINING OBJECTIVE

OmniVIVO is trained solely on speech data using cross-entropy loss:

$$\mathcal{L}_\phi = \mathbb{E}_{(x,\bar{s})} \left[ -\sum_{m=1}^{M} \log p_\phi(\bar{s}_m \mid \bar{s}_{<m}, x) \right].$$

### 3.1.5 OMNIVIVO DESIGN PROPERTIES

The OmniVIVO architecture aims to facilitate three desirable properties:

- **Preservation:** Freezing $f_\theta$ and $W_{\bar{\theta}}$ helps minimize the impact on image quality.
- **Efficiency:** Only LoRA adapters and the speech branch are trainable, reducing parameter updates.
- **Cross-modal sharing:** The backbone provides a shared semantic space between text, image, and speech.

Together, this lightweight unified design enables OmniVIVO to balance *cross-modal sharing* with *modality-specific specialization*

### 3.2 SPEECH TOKENIZER AND RECONSTRUCTION

The semantic speech tokenizer (Du et al., 2024) converts the raw speech input $S$ into intermediate feature representations $R$ through a speech encoder, SenseVoice-Large ASR model (An et al., 2024), $E_{\text{speech}}$. The encoder outputs a sequence $R = \{r_1, r_2, \ldots, r_T\}$, where $T$ is the sequence length.

$$R = E_{\text{speech}}(Y)$$

The features $R$ are quantized into discrete values using Finite Scalar Quantization (FSQ) (Mentzer et al., 2024), which maps each feature to a scalar within the range $[-L, L]$, where $L$ is the number of quantization levels:

$$R'_{\text{qt}} = \text{FSQ}(R)$$

Subsequently, discrete speech tokens $T_k$ are computed as:

$$S_k = \sum_{m=0}^{M-1} R'_{\text{qt}[k,m]} \cdot (2L+1)^m$$

Where $S_k$ is the speech token at time step $k$, which represents the discrete value corresponding to the quantized feature vector at time step $k$. The summation $\sum_{m=0}^{M-1}$ indicates that the token $S_k$ is generated by summing the quantized feature vector $R'_{\text{qt}}$ at time step $k$ and with the $m-th$ dimension.

Finally, **speech reconstruction** is performed in two stages. A flow matching model (Lipman et al.) first maps the quantized representations into a Mel-spectrogram, and a HiFi-GAN vocoder (Kong et al.) subsequently converts this Mel-spectrogram into high-fidelity, natural-sounding waveforms. This process is similarly applied to the speech token from OmniVIVO $\bar{S}$.

### 3.3 IMAGE TOKENIZER AND RECONSTRUCTION

To transform images into discrete symbols for AR modeling, we employ a VQGAN-based model (Esser et al.) consisting of an encoder, a vector quantizer, and a decoder.

Given an input image $y \in \mathbb{R}^{H \times W \times 3}$, the encoder compresses it into a latent representation

$$f = \text{Encoder}(y) \in \mathbb{R}^{h \times w \times C}, \quad h = H/p, \ w = W/p,$$

where $p$ is the downsampling factor and $C$ denotes the feature dimensionality.

The quantizer replaces each latent vector $f(i, j)$ with the closest entry from a learnable codebook $\mathcal{E} = \{e_1, \ldots, e_K\} \subset \mathbb{R}^{K \times C}$, containing $K$ prototype embeddings of dimension $C$. This assignment is defined as

$$q(i, j) = \arg \min_{k \in \{1, \ldots, K\}} \|f(i, j) - e_k\|_2^2.$$

The resulting discrete index map $q \in \{0, \ldots, K-1\}^{h \times w}$ is subsequently linearized into a sequence of $h \cdot w$ tokens that can be modeled autoregressively.

Finally, **image reconstruction** is performed by mapping the quantized embeddings back into pixel space, similarly to the image token from OmniVIVO $\bar{E}$:

$$\hat{y} = \text{Decoder}\big(e_{q(i,j)}\big).$$

### 3.4 DATA CREATION

We introduce VIVOGen, a dataset designed to advance multimodal applications in various domains, including language education and storytelling. The dataset consists of 100 high-fidelity samples of images and speech generated by OmniVIVO. Specifically, we use ChatGPT (OpenAI, 2023) to generate text inputs on diverse topics such as animals, pets, vehicles, nature, and more. Additionally, Whisper-V2 (Radford et al.) is used to remove low-quality speech samples, ensuring that the VIVO-Gen dataset maintains high intelligibility. Finally, human reviewers are involved in the final filtering stage, retaining only high-quality image and speech pairs. VIVOGen will be used in future studies focused on language education and storytelling, where the dataset is expected to have a meaningful impact on learning progress.

## 4 EXPERIMENTAL SETUP

**Architecture**. OmniVIVO unifies visual and speech generation within a single transformer backbone. It leverages a pretrained 36-layer transformer image generator with 20 attention heads and a hidden size of 1280, termed Omni-Core, with an 8-layer speech branch, totaling approximately 1 billion parameters, of which 225 million are trainable.

**Adaptation**. To enable efficient fine-tuning, we apply Low-Rank Adaptation (LoRA) (Hu et al., 2022) with rank $r = 16$ and scaling factor $\alpha = 16$, updating task-specific modules in Omni-Core and the speech branch while freezing most Omni-Core weights.

**Tokenization**: Text inputs are processed using the Flan-T5 tokenizer (Chung et al.) (vocabulary size 32,100). Speech tokens are generated via a pretrained Semantic Speech Tokenizer from CosyVoice2 (Du et al., 2024), and reconstructed using flow-matching models with HiFi-GAN vocoders (Kong et al.). Images are tokenized and reconstructed via the pretrained VQ-VAE from LlamaGen (Sun et al., 2024).

**Training:** OmniVIVO is fine-tuned on the LibriTTS (Zen et al., 2019) dataset (585 hours, multi-speaker) for 100,000 steps using the AdamW optimizer (Loshchilov & Hutter) $\beta = (0.9, 0.999)$, weight decay = 0.01, learning rate $1 \times 10^{-4}$, constant schedule. Training is conducted on a single H100 GPU for approximately two days, using a batch size of 14. The model minimizes cross-entropy loss, applies gradient clipping ($\|g\|_2 \leq 1.0$), and averages the last five checkpoints for stability.

**Evaluation** To evaluate the proposed OmniVIVO system, we first conduct an ablation study using Word Error Rate (WER) and Character Error Rate (CER) to assess the impact of model depth on intelligibility, using 1,000 test clean samples from LibriTTS (Zen et al., 2019). For this, we use Whisper-Large-V2 (Radford et al.), an Automatic Speech Recognition (ASR) model, to transcribe speech into text. Additionally, we perform subjective evaluations of speech and image quality across four metrics: Naturalness, Intelligibility, Multimodal Coherence, and Multimodal Ranking. Each subjective test involves 15 participants. For speech quality (Naturalness and Intelligibility), we compare OmniVIVO, VITS (Kim et al.), CosyVoice2 (Du et al., 2024), and Ground Truth, with each system contributing 10 samples, resulting in 40 total. For multimodal evaluations (Coherence and Ranking), only OmniVIVO is assessed, with 10 samples per experiment, as no baseline systems are available. Detailed evaluation protocols and criteria are provided in Appendix A.1.

Regarding image generation quality, we report the Inception Score (IS) Salimans et al. (2016) for OmniVIVO, LlamaGen, and recent Omni-LLM models such as MingAI AI (2025) and Unified-IO Lu et al. (2023). Note that the IS is a widely used metric that measures the image quality and diversity of generated images.

**Multimodel Ranking:** To address the lack of effective evaluation methods for multimodal generation outputs, we propose a new metric to investigate how different presentation formats influence information acquisition. We categorize the formats into three levels: **Excellent**, **Acceptable**, and **Less Effective**.

- **Excellent**: Information is conveyed quickly, clearly, and effortlessly.
- **Acceptable**: Information is sufficiently clear, though not optimal.
- **Less Effective**: Information is understandable but lacks clarity and effectiveness.

At each level, the participant will select one of the following formats: (A) Text, (B) Speech, (C) Image, (D) Text + Speech, (E) Text + Image, (F) Speech + Image, or (G) Text + Speech + Image. The results are shown in Table 5.

## 5 RESULTS

### 5.1 EFFECT OF MODEL DEPTH ON SPEECH QUALITY

Table 1: Comparison of WER and CER for TTS-Baseline and OmniVIVO across Different Model Depths. OmniVIVO achieves the best intelligibility with 8-layer

| Model Depths | TTS-Baseline | | OmniVIVO | |
|---|---|---|---|---|
| | WER↓ | CER↓ | WER↓ | CER↓ |
| 2Layer | 37.77 | 26.36 | 18.72 | 11.95 |
| 4Layer | 25.41 | 17.17 | 12.50 | 7.28 |
| 6Layer | 24.28 | 16.23 | 11.51 | 6.50 |
| 8Layer (Proposed) | 22.65 | 15.26 | **10.64** | **6.06** |
| 10Layer | 22.50 | 15.42 | 11.07 | 6.12 |

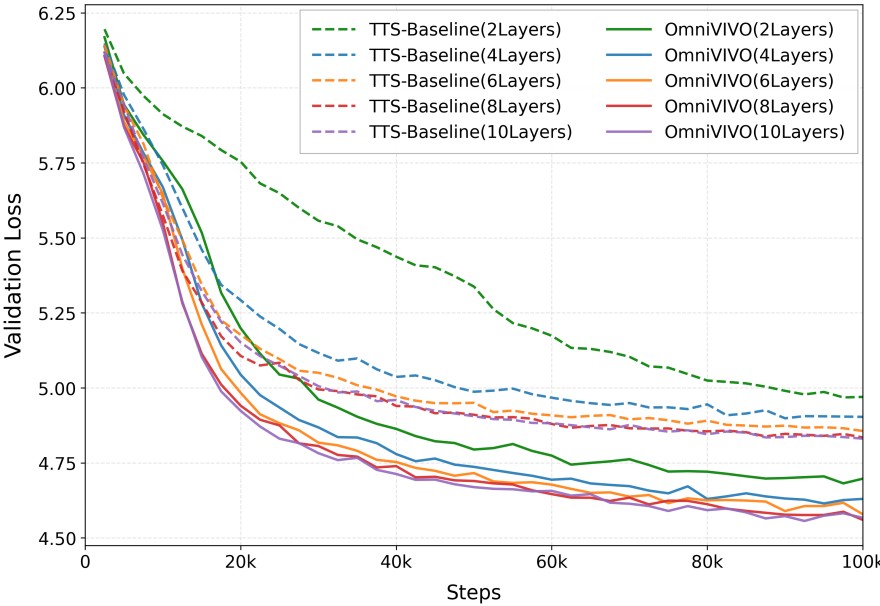

Figure 2: Comparison of validation loss for TTS-Baseline and OmniVIVO on Speech Branch across Different Model Depths.

Table 1 presents the results of an ablation study comparing OmniVIVO with a TTS baseline across five network depths (2, 4, 6, 8, and 10 layers), evaluated using WER and CER. Both models share the same architectural structure and are trained from scratch; however, OmniVIVO additionally incorporates LoRA adapters within the OmniCore to support multimodal extension (Figure 1). For evaluation, we synthesize 1,000 sentences from the LibriTTS test-clean subset and transcribe the outputs using Whisper V2. Results show that OmniVIVO consistently outperforms the baseline across all depths. At the 8-layer configuration, OmniVIVO achieves a WER of 10.64 and a CER

of 6.06, compared to the baseline's WER of 22.65 and CER of 15.26, yielding absolute reductions of 12.01 WER points and 9.20 CER points (relative reductions of 53.0% and 60.3%, respectively). Similar improvements are observed at shallower depths (e.g., 2-layer and 4-layer), demonstrating the robustness and effectiveness of the OmniVIVO design.

Furthermore, as shown in Figure 2, OmniVIVO consistently achieves lower validation loss (lower is better) compared to the TTS baseline at similar depths, reinforcing the superiority of our proposed approach.

## 5.2 IMAGE GENERATION QUALITY

Table 2: Comparison of image generation quality and diversity between OmniVIVO and omni-LLM models.

| Image Quality | Inception Score (IS) ↑ | Params |
|---|---|---|
| UnifiedIO-Base | 4.31±0.72 | 241M |
| UnifiedIO-Large | 4.51±0.72 | 776M |
| UnifiedIO-XL | 5.76±0.85 | 2.9B |
| MingOmni | 8.88±1.1 | 24B |
| LlamaGen | 7.61±0.64 | 780M |
| OmniVIVO | 7.15±1.27 | 1B |

Table 2 presents a comparison of the IS between OmniVIVO and the recent Omni-LLM model, evaluated on 250 images generated from text prompts produced by ChatGPT (OpenAI, 2023) on diverse topics. OmniVIVO achieves an IS of $7.15 \pm 1.27$, compared to LlamaGen's $7.61 \pm 0.64$. Despite having only 1B parameters, significantly smaller than MingOmni's AI (2025) 24B parameters, OmniVIVO remains competitive and even surpasses UnifiedIO-XL Lu et al. (2023). These results indicate that OmniVIVO maintains high-quality image generation without relying on large-scale models, highlighting the effectiveness of its design.

## 5.3 SUBJECTIVE SPEECH QUALITY ASSESSMENT

Table 3: Mean Opinion Scores (MOS) with 95% confidence intervals.

| Method | Naturalness↑ | Intelligibility↑ |
|---|---|---|
| GroundTruth | 4.25 ± 0.14 | 4.38 ± 0.13 |
| VITS | 3.86 ± 0.15 | 4.31 ± 0.12 |
| CosyVoice2 | 3.70 ± 0.17 | **4.33 ± 0.12** |
| OmniVIVO | **3.94 ± 0.16** | 4.19 ± 0.14 |

In this experiment, we evaluate the Mean Opinion Scores (MOS) for speech quality on two metrics, Naturalness and Intelligibility, using pretrained VITS, pretrained CosyVoice2, and our proposed OmniVIVO, as shown in Table 3. Among the generative models, OmniVIVO achieves the highest score in Naturalness ($3.94\pm0.16$), surpassing VITS ($3.86\pm0.15$) and CosyVoice2 ($3.70\pm0.17$). For Intelligibility, OmniVIVO obtains $4.19\pm0.14$, which is slightly lower than CosyVoice2 ($4.33\pm0.12$) and VITS ($4.31 \pm 0.12$), but remains competitive. These results demonstrate OmniVIVO's ability to produce highly natural-sounding and intelligible speech.

Table 4: Subjective evaluation of multimodal coherence between generated images and speech, reported with 95% confidence intervals.

| | Multimodal quality↑ |
|---|---|
| OmniVIVO | 3.79 ± 0.16 |

## 5.4 MULTIMODAL RANKING

As shown in Table 4, OmniVIVO achieves a high subjective multimodal coherence score of $3.79 \pm 0.16$. Since no prior work provides directly comparable multimodal evaluations, results are reported exclusively for OmniVIVO. Even so, the score is still informative: a value close to 4 suggests that

Table 5: Multimodal Ranking Scores for Information Acquisition Across Modalities, Unit: %

| | Information acquisition level | | |
| Modality | Excellent | Acceptable | Less Effective |
| --- | --- | --- | --- |
| Text + Speech + Image | 56.95 | 5.96 | 3.97 |
| Text + Image | 17.88 | 13.25 | 0.66 |
| Text + Speech | 9.27 | 9.27 | 1.32 |
| Speech + Image | 2.65 | 27.81 | 0.66 |
| Text | 5.96 | 15.89 | 39.74 |
| Image | 3.97 | 17.22 | 25.83 |
| Speech | 3.31 | 10.60 | 27.81 |

■ = 1st ranked, ■ = 2nd ranked, ■ = 3rd ranked.

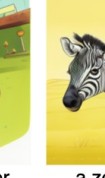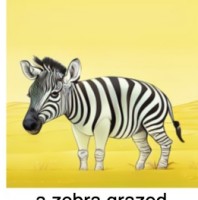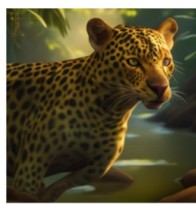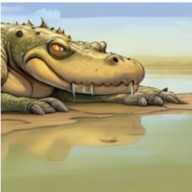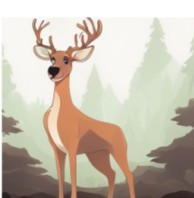

| a golden retriever chased a ball across the yard | a zebra grazed peacefully on the golden grassland | a jaguar prowled near the forest river | a crocodile basked lazily on the muddy bank | a stag stood proudly at the forest edge |

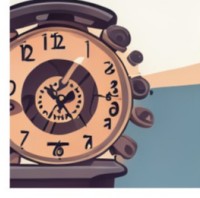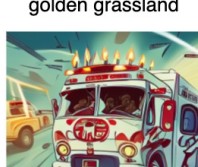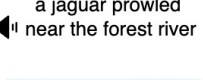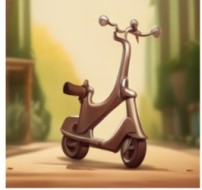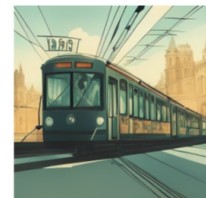

| a clock ticked loudly on the wall | an ambulance rushed through traffic with sirens loud | a tennis racket leaned against the wall | a bicycle rang its bell on the garden path | a tram rolled slowly through the city center |

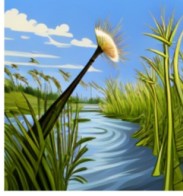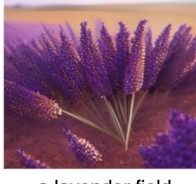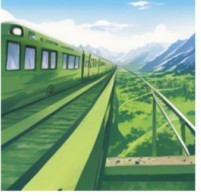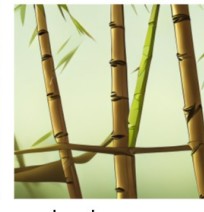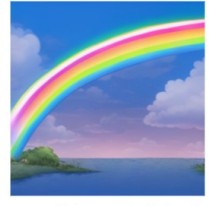

| a breeze shook the tall reeds by the river | a lavender field spread purple across the hill | a long green train rolled through the mountain valley | a bamboo grove swayed gently in the morning air | a rainbow stretched across the blue sky after rain |

Figure 3: The images and text transcripts are taken from our released dataset, VIVOGen. Note that Whisper-V2 is used to transcribe speech into text.

the generated speech and images are generally clear, aligned, and coherent. The 5-level scale used for evaluation is described in Appendix A.1.

Table 5 reports the multimodal ranking results used to assess the effectiveness of information acquisition. The combination of text input with OmniVIVO's output (speech and image) receives the highest proportion of Excellent ratings (56.95%). Dual-modality configurations (Speech + Image) are primarily categorized as Acceptable (27.81%), while single-modality options are rated as Less Effective. These results suggest that tri-modality (text + speech + image) plays an important role in improving information comprehension. Accordingly, we release VIVOGen, a high-quality dataset of 100 tri-modal samples, to support multimodal applications such as storytelling and interactive learning. Representative outputs from VIVOGen are shown in Figure 3.

Overall, these results demonstrate OmniVIVO's ability to generate coherent multimodal outputs while maintaining strong speech quality (Table 3).

## 6 CONCLUSION

In this work, we present **OmniVIVO**, the first unified autoregressive backbone capable of concurrently generating high-fidelity images and natural speech from a single text input. Unlike prior approaches that isolate modalities or combine separate generators, our OmniVIVO demonstrates the effectiveness of a single neural architecture that jointly models vision and voice within a shared multimodal space. Through extensive evaluation, we demonstrate that OmniVIVO outperforms a TTS baseline model of comparable size, and achieves comparable quality to SOTA models in both image quality (e.g., LlamaGen) and speech quality (e.g., VITS and CosyVoice2), as shown in subjective tests. Furthermore, we propose a new **multimodal ranking metric** that provides an effective way of assessing performance across modalities. Our experiments demonstrate that integrating text, image, and speech enhances information acquisition and broadens the scope of multimodal applications. In line with these findings, we target to release **VIVOGen**, a high-quality tri-modal dataset containing paired text, image, and speech data, which we expect will serve as a valuable resource for advancing multimodal generation in domains such as dynamic storytelling and education. Both the source code and dataset will be released upon acceptance of the paper.

## 7 ETHICS STATEMENT

We confirm that we have read and agree to follow the ICLR Code of Ethics. We commit to conducting our research responsibly, adhering to ethical standards throughout our involvement in the conference.

## 8 REPRODUCIBILITY STATEMENT

We confirm that our work is reproducible. We will release our source code upon acceptance of the paper.

## 9 THE USE OF LARGE LANGUAGE MODELS (LLMS)

We adhere to the ICLR guidelines regarding the use of LLMs. In this research, we used ChatGPT-5 to improve our writing and conduct surveys of related work.

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

# A  APPENDIX

## A.1  SUBJECTIVE EVALUATION METRIC

Participants rate **Naturalness** and **Intelligibility** on a 1–5 scale, with the following criteria:

**Naturalness Rating Scale**:

- **1 - Very Unnatural**: Speech sounds robotic or synthetic.
- **2 - Unnatural**: Noticeable artificiality in speech.
- **3 - Neutral**: Neither natural nor unnatural.
- **4 - Natural**: Speech is mostly natural with minor artifacts.
- **5 - Very Natural**: Fully natural, indistinguishable from human speech.

**Intelligibility Rating Scale**:

- **1 - Unintelligible**: Speech is entirely unclear.
- **2 - Poor**: Only a few words are recognizable.
- **3 - Fair**: Some segments are intelligible, but errors persist.
- **4 - Good**: Largely intelligible with minor artifacts.
- **5 - Perfect**: Fully intelligible, no effort required.

For **Multimodal Coherence**, participants evaluate the combined quality of OmniVIVO's image and speech outputs for clarity and coherence on a 1–5 scale:

**Multimodal Coherence Rating Scale**:

- **1 - Very Poor**: Image and speech are unclear, hard to understand, and not coherent.
- **2 - Poor**: Image and speech are somewhat unclear, difficult to understand, and lack coherence.
- **3 - Neutral**: Image and speech are clear but lack a seamless connection.
- **4 - Good**: Image and speech are clear, easy to understand, and mostly coherent.
- **5 - Excellent**: Image and speech are completely clear, easy to understand, and flow seamlessly.

