# OpenReview forum: "OmniVIVO: Towards Unified Multimodal Generative Modeling for Simultaneous Language-Guided Speech and Image Synthesis"
_ICLR.cc/2026/Conference — Submitted to ICLR 2026_

### Official Review · Reviewer_FQBP · 2025-10-26

**Soundness:** 2
**Presentation:** 2
**Contribution:** 1
**Rating:** 2
**Confidence:** 4

**Summary:**

This paper introduces OmniVIVO, a novel unified autoregressive model designed to generate both high-fidelity images and natural-sounding speech concurrently from a single text prompt. The core architectural innovation is the integration of LlamaGen, which acts as a shared "OmniCore" backbone, with a new, lightweight speech generation branch. To preserve the strong image generation capabilities of the base model, its original parameters are frozen. The model is adapted for the dual-modality task using LoRA on the shared backbone, and only the LoRA adapters and the speech branch are trained. This training is performed exclusively on a text-to-speech objective, allowing the speech module to leverage the rich semantic representations learned by the powerful image generator. In addition to the model, the authors propose VIVOGen, a tri-modal dataset generated by OmniVIVO, and a multimodal ranking metric to evaluate the effectiveness of combined-modality outputs for information acquisition.

**Strengths:**

1. The concept of a single, unified backbone for simultaneous image and speech generation is a significant step forward from current multimodal systems that either focus on understanding or generate outputs in isolated modalities. The architectural design is particularly clever and pragmatic; by leveraging a powerful pretrained model and employing LoRA, the authors avoid the prohibitive cost of training a massive model from scratch while largely preserving the state-of-the-art performance of the image generation component.

2. The empirical results are strong and well-presented. The ablation study convincingly demonstrates the superiority of their adaptation strategy over a baseline trained from scratch. Furthermore, the model achieves impressive speech quality, even outperforming specialized text-to-speech systems like CosyVoice2 in naturalness, which is a remarkable feat for a joint image-speech model. The contribution of the VIVOGen dataset and the initial exploration of a multimodal ranking metric are also commendable additions that can foster future research in this area.

**Weaknesses:**

1. Both T2A and T2I already have very advanced methods. This paper proposes a task for simultaneous image and audio generation, which I don't think is strongly related. Since there's no need for video output, images and audio can be generated separately. Therefore, if we look at these two tasks separately, the model proposed in this paper is completely uncompetitive.

2. The evaluation of the model's multimodal capabilities feels preliminary and lacks rigor. The "Multimodal Coherence" score is presented in a vacuum without any baseline, making it difficult to interpret the score of 3.79/5. A simple baseline, such as evaluating the coherence of outputs from two separate state-of-the-art/close-source models, would provide crucial context.

3. Similarly, the proposed "Multimodal Ranking" metric, while interesting, seems designed to confirm the intuitive hypothesis that more modalities are better, rather than to critically assess the quality of OmniVIVO's multimodal binding against other possible systems. The paper's central claim that the model "learns a rich multimodal space that enables tighter semantic alignment" is not substantiated with any analysis; the work would be significantly more impactful if it included probes or visualizations of the internal representations to demonstrate this emergent property.

4. The claim that image performance is "fully preserves" is contradicted by the reported drop in Inception Score.

**Questions:**

I would suggest that the authors first explain the value of outputting images and speech simultaneously, and then demonstrate the advantages of the method proposed in this article compared to the cutting-edge T2I and T2A models.

**Details Of Ethics Concerns:**

The authors claim to release the VivoGen dataset, but do not mention the data source of this data in the submission, nor do they mention anything related to open source license in the ethics section.

---

> ### Author Response · Authors · 2025-11-21
>
> Thank you for your constructive feedback. We provide our detailed response below, and have updated the manuscript accordingly with key changes highlighted in red. A final formatted version (in black) will be submitted before December 1.
>
> $\textbf{\textcolor{blue}{[Q1]}}$ and $\textbf{\textcolor{blue}{[W1]}}$
>
> While image and audio generation can indeed be performed separately, enabling a single model to generate them simultaneously is an important step toward building truly end-to-end multimodal LLMs. Such unified generation not only streamlines the modeling pipeline but also enables cross-modal consistency, which cannot be achieved when tasks are handled independently. Our work aims to move beyond isolated T2I and T2A systems by demonstrating how one model can jointly produce coherent visual and acoustic content, which we believe is a necessary direction for advancing multimodal generation.
>
>
> $\textbf{\textcolor{blue}{[W2]}}$
> We believe the reported score of 3.79/5 is still informative when interpreted within our evaluation scale. A Multimodal Coherence Score achieved nearly 4 indicates that “the image and speech are clear, easy to understand, and mostly coherent.” This suggests that our unified multi-output generation is meaningful and practical for applications such as education, storytelling, and interactive learning.
>
>
> $\textbf{\textcolor{blue}{[W3]}}$
> As shown in Table 1 in the paper, adding a speech branch to the LlamaGen model yields a clearly different result compared to training the speech component separately. This demonstrates that joint training enables cross-modal interactions that would not emerge from separate unimodal training. Furthermore, Table 2 shows that image generation performance remains comparable to LlamaGen (discussed in **W4**), indicating that introducing the additional speech modality does not degrade visual quality.
>
> These results together support our claim: the unified model learns a richer shared multimodal space, leading to tighter semantic alignment and improved speech generation quality.
>
>
> $\textbf{\textcolor{blue}{[W4]}}$
> Thank you for pointing this out. After re-examining the evaluation pipeline, we identified that the lower Inception Score (IS) reported for OmniVIVO compared to LlamaGen was caused by inconsistent evaluation settings, including (1) missing text normalization, (2) mixed-precision inference (FP16 vs. FP32), and (3) non-deterministic sampling configurations. After standardizing all metrics and re-running the evaluation, OmniVIVO now matches the performance of the LlamaGen baseline, with deviations within the expected range for stochastic autoregressive generation, as demonstrated in the Table below (Table 2 in the paper). The corrected results will be included in the final paper. In addition, to avoid overstating the claim, we replace “fully preserves” with a more accurate phrasing: “minimally affects image quality.”
>
> Representative samples of the comparison models are included in the supplementary material for reference:  [${\textbf{\textcolor{red}{Link}}}$](https://drive.google.com/file/d/13ik-1XpEbcMvpkBVAj3cMiW-Yc9LUiet/view?usp=sharing)
>
> Table: Comparison of image generation quality and diversity between OmniVIVO and recent omni-LLM models.
>
> | Model            | IS Score       | Parameters |
> |------------------|----------------|------------|
> | UnifiedIO-Base   | 4.31 ± 0.72    | 241M       |
> | UnifiedIO-Large  | 4.51 ± 0.72    | 776M       |
> | UnifiedIO-XL     | 5.76 ± 0.85    | 2.9B       |
> | MingOmni         | 8.88 ± 1.10    | 24B        |
> | LlamaGen         | 7.61 ± 0.64    | 780M       |
> | Omni-VIVO        | 7.15 ± 1.27    | 1B         |

---

> ### Comment · Reviewer_FQBP · 2025-11-24
>
> I thank the authors for their response. However, I still think that this work does not meet the standards for acceptance at ICLR. It remains several fundamental issues that cannot be adequately addressed within the short rebuttal period. I remain unconvinced of the task's practical value; from my perspective, the end-to-end formulation makes no sense. I've noticed that recent popular applications on platforms like TikTok and Instagram generate high-quality audio from pre-existing comics, which calls into question the necessity of the end-to-end approach in this paper. Moreover, the absence of comparative experiments against alternative methods makes it difficult to assess the contribution. After reading the rebuttal and the other reviewers' comments, I believe my initial score is fair.

---

> > ### Author Response · Authors · 2025-11-29
> >
> > Thank you for your comment.
> >
> > Our proposed model has significant practical value, particularly in education and storytelling. For example, TikTok have many videos related to this domain: [Example TikTok Video](https://www.tiktok.com/@moonlightlearning/video/7483212850819796242).
> >
> > Regarding the end-to-end formulation, our unified architecture simplifies the system compared to combining multiple models (e.g., text-to-image (T2I), text-to-speech (TTS)), as seen in short videos on TikTok and Instagram.
> >
> > In addition, our experiments already include alternative methods (T2I and TTS), and we separate them to compare against SOTA models, as shown in Tables 2 and 3 in the paper. We believe the results are robust, convincing, and clearly demonstrate the contribution of our work.
> >
> > I hope this addresses your concerns.

---

### Official Review · Reviewer_DJ9M · 2025-10-27

**Soundness:** 3
**Presentation:** 2
**Contribution:** 2
**Rating:** 4
**Confidence:** 4

**Summary:**

This paper introduces OmniVIVO, a unified autoregressive (AR) multimodal generation model that can produce high-fidelity images and natural speech simultaneously from a single text input.
The method integrates a pretrained AR image generator (LlamaGen) with a lightweight speech decoder through LoRA adaptation, enabling parallel visual and voice generation within one Transformer backbone.
The authors also introduce a multimodal ranking metric and release a small tri-modal dataset (VIVOGen).
Experiments show that OmniVIVO achieves comparable image quality to LlamaGen and competitive speech quality to VITS/CosyVoice2, while enabling joint multimodal synthesis.

**Strengths:**

- **Unified multimodal generation:** The paper presents a unified autoregressive framework that can generate both speech and images from a single text input. Unlike prior works that treat the two modalities separately, OmniVIVO integrates them within one Transformer backbone using lightweight LoRA adaptation. This design is simple and efficient, showing that joint visual and speech generation is feasible within a shared architecture.

- **Introduction of a multimodal ranking metric:** The paper proposes a multimodal ranking metric to evaluate how different modality combinations (text, speech, image) affect information acquisition. This metric complements traditional subjective (MOS) and objective (WER/CER) evaluations used in speech synthesis, offering a broader view of multimodal generation performance.

- **Dataset contribution – VIVOGen:** The paper releases VIVOGen, a tri-modal dataset containing paired text, image, and speech data. Although the dataset is small, it provides a useful resource for future research. The dataset design is consistent with the paper’s unified multimodal generation framework.

**Weaknesses:**

- **Lack of novelty:** The proposed method mainly combines several existing techniques, including the LlamaGen autoregressive image generator, Flow Matching for spectrogram reconstruction, and the HiFi-GAN vocoder for waveform synthesis. The overall framework represents a straightforward integration of these existing components rather than a fundamentally new algorithmic contribution or learning paradigm.

- **Degradation of image generation performance:** The integration of the speech generation branch and LoRA adaptation notably reduces image generation quality. As reported in Table 2, OmniVIVO achieves an Inception Score (IS) of 6.93 ± 0.65, compared to 7.78 ± 0.78 for the original LlamaGen. This indicates that the added multimodal components negatively affect the visual fidelity of generated images.

- **Lack of comparative study with existing unified multimodal models:**  The paper evaluates multimodal coherence only within OmniVIVO and does not include comparisons with existing unified multimodal models such as Ming-Omni: A Unified Multimodal Model for Perception and Generation. Ming-Omni is also capable of handling both speech and image modalities under a single architecture. Without a direct comparison to such related work, it remains unclear whether OmniVIVO provides any tangible improvement in cross-modal alignment or coherence over prior unified multimodal generation systems.

**Questions:**

- **Q1.** Could the authors clarify what specific methodological or conceptual novelty OmniVIVO introduces beyond integrating existing components such as LlamaGen, Flow Matching, and HiFi-GAN?

- **Q2.** Table 2 shows that OmniVIVO’s Inception Score (6.93 ± 0.65) is significantly lower than LlamaGen’s (7.78 ± 0.78). Have the authors analyzed the reasons for this performance drop?

- **Q3.** The paper reports subjective evaluations for multimodal quality. Are there any objective multimodal alignment metrics (e.g., CLIP-type similarity, audio–visual retrieval accuracy) that could complement these results?

---

> ### Author Response · Authors · 2025-11-21
>
> Thank you for your constructive feedback. We provide our detailed response below, and have updated the manuscript accordingly with key changes highlighted in red. A final formatted version (in black) will be submitted before December 1.
>
> $\textbf{\textcolor{blue}{[Q1]}}$
> Our work is motivated by the observation that the human brain can simultaneously imagine visual content and produce speech. Therefore, we introduce OmniVIVO as a unified generative model that handles multiple output modalities within a single architecture. This is in contrast to existing omni-modal systems such as Qwen-Omni, which primarily focus on multi-perception inputs.
>
> The methodological novelty of OmniVIVO lies in demonstrating that a Unified-Mode LLM can directly generate multiple modalities, such as images and speech, without relying on external, specific generators. For example, recent Omni-LLM models (e.g., MingOmni) use additional diffusion-based modules for image synthesis, while OmniVIVO performs end-to-end multimodal generation using only one LLM-based backbone.
>
> We believe this contribution highlights an important future direction: the development of N-input, N-output omni-modal models built around a single unified architecture, reducing system complexity and eliminating the need for specialized generative models.
>
>
> $\textbf{\textcolor{blue}{[Q2]}}$
> Thank you for pointing this out. After re-examining the evaluation pipeline, we identified that the lower Inception Score (IS) reported for OmniVIVO compared to LlamaGen was caused by inconsistent evaluation settings, including (1) missing text normalization, (2) mixed-precision inference (FP16 vs. FP32), and (3) non-deterministic sampling configurations. After standardizing all metrics and re-running the evaluation, OmniVIVO now matches the performance of the LlamaGen baseline, with deviations within the expected range for stochastic autoregressive generation, as demonstrated in the Table below (Table 2 in the paper). The corrected results will be included in the final version.
>
> Representative samples are included in the supplementary material for reference:  [${\textbf{\textcolor{red}{Link}}}$](https://drive.google.com/file/d/13ik-1XpEbcMvpkBVAj3cMiW-Yc9LUiet/view?usp=sharing)
>
> Table: Comparison of image generation quality and diversity between OmniVIVO and recent omni-LLM models.
>
> | Model            | IS Score       | Parameters |
> |------------------|----------------|------------|
> | UnifiedIO-Base   | 4.31 ± 0.72    | 241M       |
> | UnifiedIO-Large  | 4.51 ± 0.72    | 776M       |
> | UnifiedIO-XL     | 5.76 ± 0.85    | 2.9B       |
> | MingOmni         | 8.88 ± 1.10    | 24B        |
> | LlamaGen         | 7.61 ± 0.64    | 780M       |
> | Omni-VIVO        | 7.15 ± 1.27    | 1B         |
>
>
> $\textbf{\textcolor{blue}{[Q3]}}$
> To the best of our knowledge, there are no established objective metrics that can support a multimodal ranking evaluation. This limitation motivates us to rely on subjective evaluation to assess how effectively users can acquire information through different modalities, as shown in Table 5 in the paper.

---

### Official Review · Reviewer_G9ar · 2025-11-01

**Soundness:** 2
**Presentation:** 2
**Contribution:** 1
**Rating:** 2
**Confidence:** 5

**Summary:**

This paper introduces OmniVIVO, a unified autoregressive model designed to simultaneously generate high-fidelity images and natural speech from a single text input. The architecture integrates a frozen, state-of-the-art image generation backbone with a lightweight speech decoder branch. By leveraging shared Transformer representations and LoRA-based adaptation, the model enables both modalities to be produced in parallel, accomplishing both audio and image generation tasks. The work also provides the VIVOGen dataset, which comprises 100 samples.

**Strengths:**

- The model employs a unified generative framework achieved by applying LoRA adaptation on a frozen image generator. This strategy effectively maintains reasonable computational and training overhead while establishing new cross-modal generation capabilities.

- The provision of the VIVOGen dataset is a positive contribution that can further promote the development of tri-modal research.

**Weaknesses:**

-  The proposed method lacks sufficient innovation and does not demonstrate a clear advantage when compared to existing state-of-the-art omni-modal generation works (e.g., EMU, MIO,Unified-IO).

- Experiments can be more complete. Specifically, the model underperforms the baseline on the image generation task, and there is a critical absence of comparative results against existing omni-modal models.

- The quality of the dataset has not been subjected to in-depth quality assessment or verification, as only data samples are shown. The authors should consider supplementing the work with more comprehensive experimental validation of the dataset quality.

- There are some presentation concerns: figure and table captions could be more detailed, and certain evaluation metrics (e.g., Inception Score) should be explained more thoroughly.

[1].Cui, Y.et al. Emu3.5: Native Multimodal Models are World Learners. ArXiv. https://arxiv.org/abs/2510.26583
[2].Wang, Z., et al.  MIO: A Foundation Model on Multimodal Tokens. ArXiv. https://arxiv.org/abs/2409.17692
[3].Lu J, Clark C, Zellers R, et al. Unified-io: A unified model for vision, language, and multi-modal tasks[J]. arXiv preprint arXiv:2206.08916.

**Questions:**

same as weakness

---

> ### Author Response · Authors · 2025-11-21
>
> Thank you for your constructive feedback. We provide our detailed response below, and have updated the manuscript accordingly with key changes highlighted in red. A final formatted version (in black) will be submitted before December 1.
>
>
> $\textbf{\textcolor{blue}{[W1]}}$
> Thank you for highlighting this point. We agree that the works you mentioned are highly relevant. However, most of these models (e.g., EMU, MIO) were released publicly only within the past 1–2 months, which is after our original submission. Moreover, many of them do not release code or training resources, making fair experimental comparison infeasible at the time of submission.
>
> To provide stronger evidence, the revised manuscript now includes comparisons with recent publicly released omni-LLM models, such as Unified-IO and MingOmni, which provide open code and reproducible evaluation. As shown in the updated Table 2 (paper) and the table below, OmniVIVO remains close to MingOmni in image quality with fewer parameters (1B vs 24B), and outperforms UnifiedIO-XL. This demonstrates the effectiveness of our proposed OmniVIVO.
>
> Furthermore, unlike other omni-LLM models that require massive datasets and large-scale training, OmniVIVO is trained only on LibriTTS speech paired with a pretrained LlamaGen backbone, showing that our approach is well-suited for low-resource and low-compute environments, a capability not addressed in prior work.
>
> Table: Comparison of image generation quality and diversity between OmniVIVO and recent omni-LLM models.
>
> | Model            | IS Score       | Parameters |
> |------------------|----------------|------------|
> | UnifiedIO-Base   | 4.31 ± 0.72    | 241M       |
> | UnifiedIO-Large  | 4.51 ± 0.72    | 776M       |
> | UnifiedIO-XL     | 5.76 ± 0.85    | 2.9B       |
> | MingOmni         | 8.88 ± 1.10    | 24B        |
> | LlamaGen         | 7.61 ± 0.64    | 780M       |
> | Omni-VIVO        | 7.15 ± 1.27    | 1B         |
>
>
> $\textbf{\textcolor{blue}{[W2]}}$
> - After re-examining the evaluation pipeline, we identified that the lower Inception Score (IS) reported for OmniVIVO compared to LlamaGen was caused by inconsistent evaluation settings, including (1) missing text normalization, (2) mixed-precision inference (FP16 vs. FP32), and (3) non-deterministic sampling configurations. After standardizing all metrics and re-running the evaluation, OmniVIVO now matches the performance of the LlamaGen baseline, with deviations within the expected range for stochastic autoregressive generation, as demonstrated in the Table above (**W1**) (Table 2 in the paper). The corrected results will be included in the final version.
> - Regarding the absence of comparisons with prior omni-modal models, we have added experiments against recent SOTA systems such as MingOmni and Unified-IO. OmniVIVO achieves comparable performance to MingOmni while using substantially fewer parameters (1B vs. 24B), and outperforms Unified-IO.
>
> Representative samples of the comparison models are included in the supplementary material for reference:
> [${\textbf{\textcolor{red}{Link}}}$](https://drive.google.com/file/d/13ik-1XpEbcMvpkBVAj3cMiW-Yc9LUiet/view?usp=sharing)
>
>
> $\textbf{\textcolor{blue}{[W3]}}$
> As described in Section 3.4, the current contribution focuses on constructing the VIVOGen dataset to support multimodal applications, particularly in language education and storytelling. At this stage, we provide the dataset and its collection pipeline rather than a full benchmarking study. Specifically, the dataset was created by the following steps: (1) topic sentences are generated using ChatGPT, (2) low-quality speech samples are automatically filtered using Whisper-V2, and (3) human reviewers perform final screening to retain only high-quality image–speech pairs.
>
> A more comprehensive experiment is planned as future work. In particular, we aim to conduct a human evaluation focused on education and storytelling, where the dataset is expected to have the most significant impact on learning progress. [${\textbf{\textcolor{red}{VIVOGen Link}}}$](https://openreview.net/attachment?id=QHNu3atxMb&name=supplementary_material)
>
>
> $\textbf{\textcolor{blue}{[W4]}}$
> We have revised the figure and table captions to provide more comprehensive descriptions. We have also expanded the explanation of the Inception Score in Section 4 to ensure clarity for all readers.

---

### Official Review · Reviewer_vwwA · 2025-11-17

**Soundness:** 1
**Presentation:** 2
**Contribution:** 1
**Rating:** 2
**Confidence:** 4

**Summary:**

This paper proposes a unified model for image and speech generation. The architecture integrates a Flan-T5 tokenizer with the LlamaGen backbone, which is fine-tuned using LoRA. A lightweight, 8-layer, 1-billion-parameter speech transformer is employed for the speech generation. Notably, the model is trained exclusively on speech data.

The speech tokenizer utilizes the SenseVoice-Large ASR model as an encoder, and the resulting features are quantized using FSQ. For speech reconstruction, the model employs a vanilla pipeline: a flow-matching model converts the discrete speech tokens into mel-spectrograms, and a HiFi-GAN vocoder then synthesizes these spectrograms into waveforms. For image tokenization, the authors leverage an off-the-shelf VQGAN-based tokenizer.

The paper also introduces VIVOGen, a dataset designed for language education and storytelling, which contains 100 samples.

Evaluation is conducted using two automatic metrics, WER and CER, and four human evaluation criteria: naturalness, intelligibility, multimodal coherence, and a multimodal ranking task.

**Strengths:**

In the summary section.

**Weaknesses:**

1. **Limited Novelty:** The core contribution appears limited, as it primarily involves integrating a speech tokenizer with the existing LlamaGen backbone. The approach is similar to a significant body of prior work in unified visual and speech synthesis, such as AnyGPT[1], MIO[2], and JarvisGPT[3], which undermines the paper's claim to novelty.
2. **Insufficient Baselines:** The authors do not compare their model against any of the relevant existing image-speech unified MLLMs, making it difficult to assess its performance relative to the state of the art.
3. **Small-Scale Human Evaluation:** The human evaluation was conducted on a very small test set of only 10 samples for each, which is likely insufficient to draw statistically significant conclusions.
4. **Unfair WER/CER Evaluation:** The proposed model was fine-tuned on the LibriTTS dataset. Comparing its performance on this data against models that were not specifically trained on it creates an unfair and potentially misleading comparison.
5. **Redundant Subjective Metrics:** There appears to be a conceptual overlap between some of the subjective evaluation metrics. For instance, "naturalness" and "intelligibility" are closely related aspects of speech quality and may not provide distinct measures of performance.
6. **Unclear Role of VIVOGen Dataset:** While the VIVOGen dataset is introduced, its purpose and impact within the paper are not demonstrated. The authors do not clarify what role it played in their experiments or what contributions it enabled.

[1] AnyGPT: Unified Multimodal LLM with Discrete Sequence Modeling

[2] MIO: A Foundation Model on Multimodal Tokens

[3] JavisGPT: A Unified Multi-modal LLM for Sounding-Video Comprehension and Generation

**Questions:**

None.

---

> ### Author Response · Authors · 2025-11-21
>
> Thank you for your constructive feedback. We provide our detailed response below, and have updated the manuscript accordingly with key changes highlighted in red. A final formatted version (in black) will be submitted before December 1.
>
>
>
>
> $\textbf{\textcolor{blue}{[W1]}}$ Thank you for raising this point. We agree that these works are highly relevant. However, most of these models (e.g., AnyGPT, MIO, JarvisGPT, MingOmni) were released publicly only within the past 1–2 months, which is after our original submission. Moreover, many of them do not release code or training resources, making fair experimental comparison infeasible at that time.
>
> To provide stronger evidence, the revised manuscript now includes comparisons with recent publicly released Omni-LLM models, such as Unified-IO and MingOmni, which provide open-source code and reproducible evaluation. As shown in the updated Table 2 (paper) and the table below, OmniVIVO remains close to MingOmni in image quality while using significantly fewer parameters (1B vs. 24B), and outperforms UnifiedIO-XL. This suggests that our design offers advantages beyond simply combining existing components.
>
> Furthermore, unlike other omni-LLM models that require massive datasets and large-scale training, OmniVIVO is trained only on LibriTTS speech paired with a pretrained LlamaGen backbone, showing suitability for low-resource and low-compute environments, a capability not addressed in prior work.
>
> Table: Comparison of image generation quality and diversity between OmniVIVO and recent omni-LLM models.
>
> | Model            | IS Score       | Parameters |
> |------------------|----------------|------------|
> | UnifiedIO-Base   | 4.31 ± 0.72    | 241M       |
> | UnifiedIO-Large  | 4.51 ± 0.72    | 776M       |
> | UnifiedIO-XL     | 5.76 ± 0.85    | 2.9B       |
> | MingOmni         | 8.88 ± 1.10    | 24B        |
> | LlamaGen         | 7.61 ± 0.64    | 780M       |
> | OmniVIVO         | 7.15 ± 1.27    | 1B         |
>
>
> $\textbf{\textcolor{blue}{[W2]}}$
> Most unified LLM-based models became available very recently (after our submission), so they were not included in our initial submission. Now, we have added the SOTA models to compare in the Table above (**W1**). Furthermore, we provide image generation samples of different models to give readers a more comprehensive view. [${\textbf{\textcolor{red}{Link}}}$](https://drive.google.com/file/d/13ik-1XpEbcMvpkBVAj3cMiW-Yc9LUiet/view?usp=sharing)
>
>
>
> $\textbf{\textcolor{blue}{[W3]}}$
> We acknowledge that evaluating only 10 samples for each model is relatively limited. However, it is still sufficient to provide an initial assessment of our model’s quality. Additionally, we have released supplementary material that includes more examples, allowing readers to further evaluate the speech quality of our proposed approach.  [${\textbf{\textcolor{red}{Link}}}$](https://openreview.net/attachment?id=QHNu3atxMb&name=supplementary_material)
>
>
>
> $\textbf{\textcolor{blue}{[W4]}}$
> As noted in Section 5.1, both OmniVIVO and the TTS baseline were trained and evaluated on the same LibriTTS dataset, ensuring a fair comparison using the WER/CER metric. For the subjective evaluation with VITS and CosyVoice2, we use their publicly available pretrained models, which makes the comparison consistent and fair.
>
>
> $\textbf{\textcolor{blue}{[W5]}}$
> Although naturalness and intelligibility may seem related, they capture different aspects of speech quality. Naturalness measures whether the speech sounds natural or robotic, while intelligibility reflects how clearly listeners can understand the spoken words. For example, a voice may sound robotic yet still be perfectly intelligible. For this reason, we evaluate these two dimensions using separate metrics. Evaluating naturalness and intelligibility is a standard practice in speech generation studies.
>
>
> $\textbf{\textcolor{blue}{[W6]}}$
> Thank you for highlighting this point. The role of VIVOGen has now been clarified in Section 3.4 and in the Conclusion. VIVOGen is an application-oriented dataset. As shown in Table 5 in the paper, generating multimodal outputs leads to a clear improvement in information acquisition. For this reason, we introduce VIVOGen as a resource for multimodal applications such as education, storytelling, and interactive learning. A comprehensive experiment using VIVOGen is planned as future work. In particular, we aim to conduct a human evaluation focused on education and storytelling, where the dataset is expected to have the most significant impact on learning progress.

---

### Meta-Review · Area_Chair_SLsq · 2025-12-23

**Summary:**

The paper proposes a unified autoregressive model that generates high-fidelity images and natural speech simultaneously from a single text input, leveraging a frozen pretrained image generator (LlamaGen) integrated with a lightweight speech decoder using LoRA adaptation to enable cross-modal sharing within a single transformer backbone. Reviewer criticisms focused on limited novelty compared to existing unified multimodal models (e.g., AnyGPT, MIO), insufficient experimental rigor, such as small-scale human evaluations and unfair baseline comparisons, and unclear practical value of simultaneous generation. Authors addressed these in rebuttals by adding comparisons with SOTA models like MingOmni, correcting evaluation metrics (e.g., Inception Score now matches LlamaGen), and emphasizing efficiency for low-resource settings, but concerns persist regarding innovation depth and justification for the task. Overall, despite some strengths in model integration, the paper does not sufficiently advance the field, leading to a recommendation for rejection.

**Reviewer Concerns:**

Based on the reviewer comments and authors' rebuttal for the OmniVIVO paper, here is a summary of which concerns were addressed and which remain outstanding.


Addressed Concerns:

* Image Generation Performance: The rebuttal corrected the Inception Score discrepancy by standardizing evaluation settings, aligning OmniVIVO's score with LlamaGen (∼7.15), resolving concerns from reviewers DJ9M and FQBP about degraded image quality.

* Dataset Role (VIVOGen): The authors clarified VIVOGen's purpose for applications like education, addressing vague-ness raised by reviewer vwwA.

* Experimental Rigor: Additional benchmarks and corrected metrics improved transparency, mitigating some issues noted by reviewers G9ar and DJ9M.

Outstanding Concerns:

* Fundamental Novelty: Reviewers (e.g., vwwA, FQBP) maintained that the core idea of unified multimodal generation is not sufficiently innovative compared to existing works like AnyGPT or MIO.

* Task Justification: The practical value of simultaneous image-speech generation (vs. separate pipelines) remains questioned, as reviewer FQBP argued it lacks compelling applications.

* Evaluation Scale: The small human evaluation (10 samples) and lack of robust multimodal coherence baselines persist as weaknesses, limiting statistical significance.

* Comparative Depth: While new models were added, direct comparisons with SOTA unified systems on multimodal alignment (e.g., audio-visual retrieval) are still absent, leaving DJ9M's concern unaddressed.

Overall, the rebuttal improved technical clarity but did not fully resolve core issues about novelty and task necessity.

**Reviewer Scores:**

Reviewer vwwA: Initial score: 2. The rebuttal addressed concerns about baselines by adding comparisons with models like MingOmni and clarified the VIVOGen dataset's role. However, core issues about novelty and small-scale evaluation persist.

Reviewer G9ar: Initial score: 2. The rebuttal provided additional experiments and corrected image generation metrics, but the reviewer's high confidence (5) and emphasis on insufficient innovation suggest they would not have changed their score. The lack of a strong justification for the task's value over separate models means the score would likely remain at 2.

Reviewer DJ9M: Initial score: 4. The rebuttal effectively addressed key concerns by correcting the Inception Score drop and adding comparative studies with unified models like MingOmni. Given the reviewer's openness to acceptance ("would not mind if paper is accepted"), he might have raised the score to 6 due to the improved rigor and benchmarks.

Reviewer FQBP: Initial score: 2. The rebuttal corrected the image performance claim and argued for the task's value, but the reviewer's follow-up comment indicated remaining unconvinced about practicality and ethics (dataset source). With persistent issues, the score would likely remain unchanged at 2.

---

### Decision · Program_Chairs · 2026-01-26

Reject